# Life-History and Ecological Correlates of Egg and Clutch Mass Variation in Sympatric Bird Species at High Altitude

**DOI:** 10.3390/biology12101303

**Published:** 2023-10-02

**Authors:** Yuxin Liu, Xiaolong Du, Guopan Li, Yingbao Liu, Shaobin Li

**Affiliations:** College of Life Sciences, Yangtze University, Jingzhou 434025, China

**Keywords:** breeding biology, phylogenetic analysis, reproduction, Tibetan Plateau

## Abstract

**Simple Summary:**

This study intended to understand the variation in egg and clutch mass among coexisting bird species at high altitudes. We investigated several life-history or ecological factors that could explain this variation. The results show that both egg and clutch mass were related to body mass across species. Contrary to the hypothesis of a trade-off between egg mass and clutch size, egg mass variation was not explained by clutch size when accounting for allometric effects. Clutch mass was found to be positively associated with parental care and negatively associated with predation rate. When considering clutch size and egg mass together, clutch size was significantly correlated with parental care, predation rate, and lifespan, while egg mass was only associated with development period. These results support the idea that reduced clutch size or mass is linked to a higher risk of predation, reduced parental care, and longer adult lifespan. Our findings suggest that clutch size has a greater impact on these factors compared to egg mass, possibly because smaller clutches represent a more significant reduction in energetic investment. This study has increased our understanding of how different factors influence the size of eggs and clutches in coexisting high-altitude bird species.

**Abstract:**

The variation in egg and clutch mass in sympatric species at high altitudes is poorly understood, and the potential causes of variation are rarely investigated. This study aimed to describe the interspecific variation in avian egg and clutch mass among 22 sympatric bird species at an altitude of 3430 m. Our objective was to reduce potential confounding effects of biotic/abiotic factors and investigated hypotheses concerning allometry, clutch size, parental care, nest predation, and lifespan as possible correlates and explanations for the observed variation. Our findings indicated that both egg and clutch mass evolve with body mass across species. We found that egg mass variation was not explained by clutch size when controlling for allometric effects, which contrasts the “egg mass vs. clutch size trade-off” hypothesis. Additionally, we found that clutch mass was positively associated with parental care (reflected by development period) but negatively associated with predation rate. By substituting egg mass and clutch size into the models, we found that clutch size was significantly correlated with parental care, predation rate, and lifespan, while egg mass was only significantly associated with development period. Overall, these findings support life-history theories suggesting that reduced clutch size or mass is associated with a higher risk of predation, reduced parental care, but longer adult lifespan. Interestingly, our results indicate that clutch size has a greater influence on these factors compared to egg mass. This could be attributed to the fact that smaller clutch sizes result in a more notable decrease in energetic allocation, as they require a reduced effort in terms of offspring production, incubation, and feeding, as opposed to solely reducing egg size. These findings contribute to the growing evidence that life-history and ecological traits correlate with egg and clutch mass variation in sympatric species. However, further research is needed to explore the potential evolutionary causes underlying these patterns.

## 1. Introduction

Egg mass varies across species and is related to body mass in various taxa [1,2,3]. However, body mass only accounts for part of the variation in egg mass, and the factors contributing to the remaining unexplained variation are still poorly understood. A fundamental principle in life-history theory is the existence of a trade-off between egg size and clutch size [4]. Notably, high-altitude species tend to exhibit smaller clutch sizes but larger eggs compared to their low-altitude counterparts [5,6]. Nevertheless, it is widely assumed that this trade-off arises from the allocation of limited resources between the number and mass of eggs [4,6]. Positive covariation of resource-based traits, such as egg mass and number, may be expected if species vary in their resource acquisition skills [7,8,9]. Furthermore, it is important to consider that selection processes may influence the overall reproductive investment and potentially obscure the trade-offs between egg mass and clutch size [10,11]. Although previous investigations have explored the variation in egg mass and its connection to life-history traits in avian species, the interspecific trade-offs between egg mass and clutch size remain a subject of debate due to potential confounding factors [12,13]. More interspecific studies with reduced confounding factors are still needed to investigate whether such trade-offs exist [14].

Parental care plays a crucial role in the allocation of reproductive investment in avian species, and the selection for enhanced reproductive investment may lead to correlated selection on various components of reproductive investment [10,15,16,17]. The allocation of resources towards reproduction can potentially result in a positive correlation between reproductive investment and both egg and clutch mass [4,14]. However, it is important to note that such correlations are not invariably observed [1,17,18]. In this regard, clutch mass, which encompasses both the energetic expenditure for egg production and the number of offspring necessitating parental care, may offer a more comprehensive indicator of investment in egg production compared to solely examining egg mass or clutch size. Furthermore, the development period from incubation to fledging varies among atricial species and implies effort of parental care [19]. Therefore, it is necessary to investigate whether heavier egg or clutch mass are associated with greater parental care reflected by longer development periods across avian species.

Nest predation is the primary cause of nest failure in avian species [4,20], and an escalation in egg predation may directly lead to reduced reproductive investment, potentially manifesting as diminished clutch size or clutch mass [21,22,23]. However, if a trade-off exists between clutch size and egg mass, and nest predation influences clutch size, it is plausible that nest predation indirectly affects egg mass by directly impacting clutch size. Nevertheless, investigations into the effects of predation risk on variations in egg and clutch mass remain scarce [23]. Furthermore, it is anticipated that reproductive effort decreases as adult survival or lifespan increases [4,19,24]. Consequently, species characterized by longer lifespans may exhibit a preference for allocating fewer overall resources through reduced clutch size or clutch mass. However, they may compensate for this by investing more heavily in individual offspring through larger egg mass, thereby potentially enhancing offspring quality and promoting improved longevity [11,25]. To elucidate further, it is plausible that clutch size and clutch mass exhibit a positive relationship, while egg mass demonstrates a positive association with adult survival, not solely due to a trade-off between clutch size and egg mass, but rather as a result of correlated selection stemming from increased lifespan. However, these theoretical predictions have received limited empirical investigation.

Earlier studies primarily investigated variations in egg mass and clutch mass based on species inhabiting different low-land regions [14], while high-altitude avian communities are understudied in terms of egg and clutch mass variation [26,27]. High altitudes are associated with adverse and challenging climate, and the breeding season is much shorter for avian species there. Previous interspecific analysis seldom collected data at extremely high altitudes (>3400 m), where short breeding season only allows a single brood per year for most avian species as it is near the up limit of their altitudinal distribution [26,28]. The confounding effect—number of yearly broods, which is hard to control for species in low-land studies, can be easily reduced at high altitudes. Besides, the habitat at high altitudes tends to be more homogenous [29,30,31], while diverse habitat at low land might confound the final results. Species of different low-land regions experiences rather different biotic and abiotic environment (e.g., climate, vegetation and predator density) which might confound the final results [14]. Therefore, an interspecific analysis of sympatric species at extremely high altitude can provide an appealing opportunity to examine egg and clutch mass variation while controlling for various confounding effects of biotic/abiotic factors.

In this study, we collected data on sympatric bird species at an altitude above the tree line to minimize the confounding effects of biotic/abiotic factors, and conducted phylogenetic-controlled analyses to investigate the interspecific variation in egg/clutch mass and its life-history and ecological correlates near the upper limit of their breeding distribution across avian species which was rarely included in former studies. Life-history traits, such as clutch size and lifespan, as well as ecological factors like nest predation, may directly influence the evolutionary process of egg and clutch mass. However, they may also exert an indirect influence through trade-offs in overall reproductive tactics. Therefore, the aims of this study are to comprehensively describe the variation of egg/clutch mass through allometric relationship, and to investigate potential life history or ecological correlates of clutch/egg mass variation at interspecific level in sympatric area.

## 2. Study Site and Methods

The study site was located at Tianjun Meadow in the northeast of the Tibetan Plateau (37°17′ N, 99°06′ E, 3400 m in altitude). The meadow is primarily used for grazing of livestock and is intersected by two intermittent streams that originate from the southern mountains. These streams typically dry up until the rainy season, which usually occurs in June. The study area covers approximately 600 ha of publicly accessible grassland, primarily characterized by alpine steppe meadow vegetation. The average annual temperature recorded in this area was −1.0 °C with a total annual rainfall of 350 mm based on data collected from 1990 to 2010. Around 30 bird species nest in this area, and further details on the research site are available in Li et al. [29].

Fieldwork was conducted during the breeding seasons between 2009 and 2018. From May to August, an extensive survey of bird nests was conducted within the study area. The identification of nest location was accomplished through flushing the incubating individuals or by closely tracking adults carrying nest materials and food during the incubation or nestling periods, ultimately leading to the discovery of nests. Once a nest was located, its geographic location was recorded with a GPS receiver, and it was monitored every 2–4 days to obtain data on egg mass, clutch size, mass of the clutch, incubation period, nestling stage, and nest fate (if any). Direct inspection was performed on open-cup nests, whereas cavity nests were assessed using either a miniature camera mounted on a pole or by excavating a hole at the side of the nest chamber. During the incubation or nestling period, we conducted minor excavation on certain cavity nests that featured long tunnels. A vertical shaft was carefully dug in close proximity to the cavity chamber, allowing us to examine the contents of the nest. To facilitate future potential inspections, the shaft was effectively sealed using bags filled with soil and covered with original turfs. No apparent negative impacts were observed for this method [29,32,33]. As the nests approached hatching or fledging, the frequency of inspections was intensified to occur every 1–2 days. Adults were captured using mist nets, which were complemented by the playback of conspecific songs. Alternatively, rope traps were employed, consisting of two intersecting strings positioned above the nest, and each string was equipped with six adjustable loops crafted from horsetail hair [5]. Adults were weighed via an electronic balance to 0.01 g. More information on field procedures are available in Li and Lu [34] and Li et al. [28,29].

Egg mass was measured before clutch completion with electronic balances to 0.01 g. Egg mass was averaged in each clutch. Clutch size was determined as the final number of eggs at two different days. For each species, clutch mass is calculated as egg mass multiply clutch size. The incubation period was defined as the duration from the initiation of incubation until the emergence of the first hatchling. Subsequently, the nestling period was defined as the time span from the first hatchling’s emergence to the departure of the first fledgling [5]. The development time was calculated by adding the incubation period and the nestling period. Nests were considered successful if they produced at least one young. Nest failure was determined in cases where the nest, eggs, or nestlings were no longer present or unattended at a stage too early for fledging to have occurred.

In this study, all the data were obtained through fieldwork conducted during the study or from literature published by our research group in the study site. The collected datasets comprised several life-history and ecological traits (e.g., body mas, egg mass, clutch size, period of incubation, clutch mass, period of nestling stage and nest fate). We use maximum longevity as a proxy of lifespan of species. Data on maximum longevity of each species came from Myhrvold et al. [35] and the AnAge database [36]. For some species (*n* = 10) whose maximum longevity is unavailable in published sources, we estimated their values by the averaged values of closely related congeners. Finally, a total of 22 species with complete datasets were collected.

## 3. Statistical Analyses

The primary cause of nest failure in bird species is predation [4,28], so we used the rate of nest failure (one minus nest success) as a proxy for nest predation rate. Prior to conducting the analyses, we assigned each species a qualitative rank score to evaluate its quality of the data. The scores ranged from low (1) to medium (2) to high (3), and were determined based on an overall assessment of factors such as sample size and the comprehensiveness of behavioral observations conducted [26,37]. To account for the varying data quality, all statistical analyses were conducted using the assigned data score as a weight variable. Comparisons were made between the results obtained with the data score and those obtained without it. While the estimated sizes were similar in both cases, the errors around these estimates and the associated *p*-values were larger when the data score was not accounted for.

Data of these species are often non-independent due to their common phylogenetic relationship [38,39,40]. We found strong phylogenetic signals for the traits examined in our study (e.g., λ = 1.06 for body mass, λ = 0.94 for clutch size, λ = 1.07 for egg mass, λ = 0.98 for predation rate, λ = 1.09 for development period, etc.). All the λ values of these traits differed significantly from 0 with *p*-values < 0.01), suggesting our models should control for phylogenetic relationship. Therefore, we employed Phylogenetic Generalized Least Squares (PGLS) models with a maximum likelihood estimation of Pagel’s λ for phylogenetic dependence that controlled for phylogenetic relationship of study species. We downloaded 1000 trees for the 22 studied species from BirdTree [41] using the Hackett et al. [42] backbone, and constructed the maximum clade credibility tree (summary tree) with the R package *phangorn* [43] to account for phylogenetic relationship among species in the subsequent analyses. Before analyses, data of predation rate was log + 1 transformed while other data were log-transformed to achieve normality [44].

Initially, we conducted a PGLS model to test whether egg mass (response variable) is associated with clutch size (predictor variable) after controlling for body mass. This model aimed to determine whether there is a trade-off between egg mass and clutch size. Then, we conducted subsequent PGLS models to test the relationship between clutch mass and each life-history traits or ecological factors; in these models, we used clutch mass as the response variable, each life-history trait or ecological factor (development period, predation rate and max lifespan) as a predictor variable, and body mass as a covariate all the PGLS models. To complement the above models, we conducted three more PGLS models to test whether each life-history trait or ecological factor (development time, predation rate, or maximum longevity) as the response variable is correlated with egg mass and clutch size (by replacing clutch mass) as the predictor variables, while controlling for body mass as a covariate.

All the analyses were conducted in R version 4.1 [45], using the package *ape*, to construct PGLS models [46]. All the PGLS models were performed with data score as a weight variable through the summary tree to account for phylogenetic relationship. We reported the mean estimates and two-tailed *p*-values for the explanatory variables in all the PGLS models. A significance threshold of 0.05 was used throughout the analysis.

## 4. Results

We collected data on egg mass, clutch size, clutch mass, several life-history traits and ecological factors for 22 sympatric species (Appendix A). Egg mass was negatively correlated with clutch size (slope = −1.39 ± 0.35 t = −3.96, *p* < 0.001), but this relationship disappeared when body mass was included as a covariate to control for allometric effect in PGLS models (Table 1; Figure 1a). This suggests that egg mass did not covary with clutch size when body mass was included as a covariate. Body mass included as a covariate is strongly associated with egg mass (Table 1) and clutch mass in all the PGLS models across species (Table 2), separately. The slope of the log–log regression between egg mass and body mass is 0.86 as the ^6^/_7_ exponent that often characterizes metabolic processes scaling with body mass [14]. The slope of the log–log regression between clutch mass and body mass is 0.73 as the ^3^/_4_ exponent that often characterizes metabolic processes scaling with body mass.

Our PGLS results revealed that clutch mass positively associated with development period when controlling for a significantly effect of body mass (Table 2, Figure 1b). Body mass as a covariate significantly increased with clutch mass in all PGLS models of Table 2 (all *p* < 0.001). Clutch mass was shown to be negatively related to nest predation rate when body mass was included as a covariate (Table 2, Figure 1c). However, clutch mass did not vary with lifespan because of a non-significant though positive slope (Figure 1d). When both clutch size and egg mass (when replacing clutch mass) were included with body mass as covariate to control for allometric effect in PGLS models, we found that clutch size has significant effect in all the models (Table 3). Clutch size increased as the development period prolonged but decreased as predation rate or lifespan increased. On the other hand, egg mass only significantly correlated with parental care reflected by development period, while the relationships between egg mass and predation rate or lifespan were absent (Table 3).

## 5. Discussion

Our results reveal that egg mass and clutch mass are strongly correlated with body mass, consistent with previous studies across various taxa [2,3,13]. This relationship remains significant when other life-history and ecological traits were included (Table 2; Figure 1), indicating that egg and clutch mass evolve similarly with body mass and are associated with life-history and ecological correlates. Our findings also show that absolute egg mass decreases with increasing clutch size across coexisting species, as predicted by theoretical models [47] and observed in other studies [12]. However, this relationship disappears when controlling for body mass, suggesting that the trade-off between clutch size and egg mass may be more complex than previously thought [13,48]. Previous studies have proposed that the inverse relationship between number and mass is due to finite resource allocation [7]. However, our findings suggest that egg production represents a minor fraction of the energy investment made by altricial birds, as they primarily allocate a substantial amount of energy towards parental care [4]. Consequently, the lack of a significant relationship between clutch size and egg mass may be attributed to intricate selection pressures influencing these two interconnected life-history traits [2,49].

Furthermore, our results provided evidence for the coevolution of egg and clutch mass alongside other life-history strategies, as demonstrated by the evolution of larger eggs in species characterized by longer developmental periods, as documented in other studies [4,26,50]. Larger eggs and clutches typically require more energy for incubation and development, resulting in longer development times. Additionally, species with larger eggs tend to have slower paces of life, reduced development rates, and longer lifespans [11,50,51]. Therefore, increasing development period may be a general requirement for larger egg and clutch mass within and across altricial bird species.

Our study revealed a significant association between increased nest predation and reduced reproductive investment, as indicated by both clutch size and clutch mass, which aligns with previous predictions [19,22]. However, the impact of nest predation on egg mass was less pronounced (Table 2). It is widely observed that larger clutch size is associated with reduced nest predation [4,52]. In our study area, most bird species only produce a single brood per year [5,29,33,53], and our previous study revealed a clear connection between reduced nest predation risk and large clutch size [26,34]. However, some studies suggest that birds reduce egg mass as a response to heightened nest predation risk [14]. The missing association between nest predation and egg mass raises questions on the significance of nest predation in shaping egg mass variation within coexisting habitat. Testing the direct effects of nest predation favoring reduced egg mass is challenging due to potential conflicting direct and indirect impacts. Nest predation may directly promote reduced investment in the current clutch through decreased egg mass [23], but indirectly favor larger eggs by directly favoring smaller clutch sizes (see Section 4) [22,25] that correlated with larger eggs. Consequently, the indirect effect of nest predation promoting smaller clutches (associated with larger eggs) opposes direct effects promoting smaller eggs, thereby increasing the complexity to detect the effects of nest predation. Further studies involving diverse bird species with varying nest predation rates are required to elucidate the potential contribution of nest predation to the variation in egg mass.

Life-history theory suggests species with longer lifespans or higher adult survival rates are expected to exhibit reduced reproductive effort [4,19,24]. Our results tend to support this prediction in that lifespan was negatively associated with reproductive effort reflected by clutch size, although negative relationship between lifespan and clutch mass was not significant. Additionally, our study did not identify a positive correlation between egg mass and lifespan, which does not to support the notion that species with longer lifespan exhibit a greater investment in individual progeny through the production of larger eggs. This is probably due to the limited sample size of species. Additional studies across more sympatric species are needed to investigate the relationship between lifespan and egg or clutch mass. Maximum longevity is another issue that has been argued to be problematic in this study, as it represents the longest-lived known individual and is therefore highly sensitive to sample size [11,54,55]. Averaged lifespan or life expectancy is a more appropriate measure in future analyses.

It has been suggested that research activities may have potential adverse impacts on avian species [4,56,57]. Investigations involving birds encompass a range of actions, including regular visits to nests and handling of the birds. These research endeavors have been linked to possible detrimental consequences for the breeding success of birds. Factors contributing to this concern include the disturbance of vegetation, which could attract predators, as well as the introduction of human odor or other cues to the nests. However, the existence of this adverse effect remains uncertain. Although we acknowledge the potential for negative consequences, previous studies have indicated that the impact of such visitation is likely to be minimal or even negligible [29,58]. In this study, we have taken measures to minimize unnecessary nest visits and, when necessary, observe the nests from a distance. Additionally, we have made diligent efforts to recover the nest site characteristics to their original condition after the monitoring process. In conclusion, our study found significant associations between egg mass, clutch size and clutch mass, and a few life-history traits and ecological factors. However, further examination of possible causes is needed by characterizing the egg and clutch mass variations across more sympatric species. A portion of the variation observed can be attributed to the evolutionary changes in life-history traits and parental care behaviors, while another portion is linked to adult survival rates and reproductive investment. Further investigation is warranted to explore the specific roles of clutch size in relation to lifespan, as well as the potential impacts of nest predation. Additionally, the effects of egg mass on embryo development and long-term fitness necessitate further examination. Our study was limited to a sample of 22 bird altricial species. Therefore, future long-term fieldwork and well-designed experiments including both altricial and precocial species would be particularly beneficial in elucidating the life-history and ecological associations of egg and clutch mass.

## 6. Conclusions

This study investigated the variation in egg and clutch mass among coexisting bird species at high altitude, and explored several life-history/ecological factors that may explain this variation. We found that both egg and clutch mass were correlated with body mass across species. Contrary to the hypothesis of a trade-off between egg mass and clutch size, we found that egg mass variation was not influenced by clutch size when accounting for allometric effects. Furthermore, clutch mass was positively associated with parental care and negatively associated with predation rate. We observed that clutch size was significantly correlated with parental care, predation rate, and lifespan, whereas egg mass was only associated with development period. These results support the notion that reduced clutch size or mass is linked to a higher risk of predation, reduced parental care, and longer adult lifespan. Importantly, our findings suggest that clutch size has a greater impact on these factors compared to egg mass, possibly due to the significant reduction in energetic investment associated with smaller clutches. Overall, this study enhances our understanding of the factors influencing the size of eggs and clutches in coexisting high-altitude bird species. Further study is needed to explore the evolutionary causes underlying these patterns.

## Figures and Tables

**Figure 1 biology-12-01303-f001:**
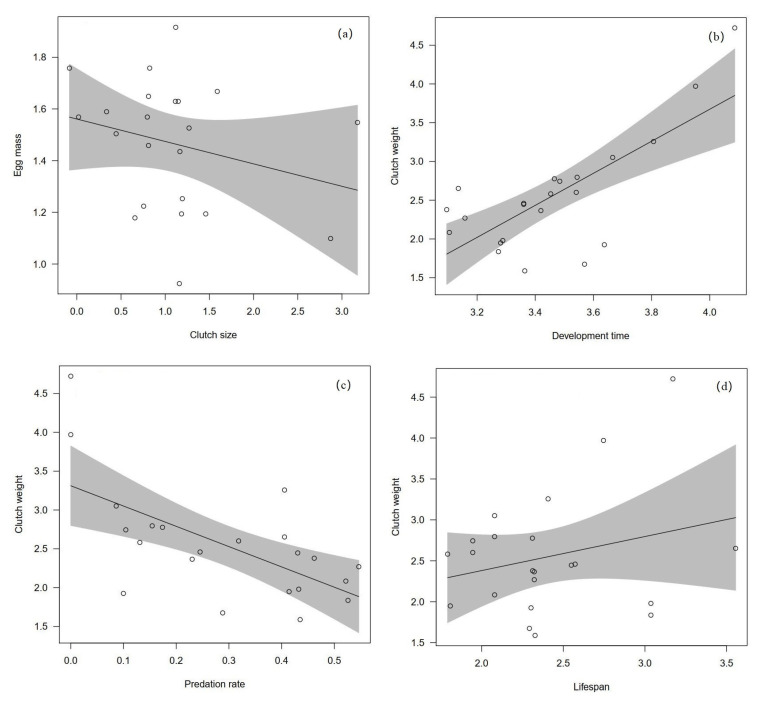
Association between of egg mass and clutch size (**a**), clutch weight and development period (**b**), clutch weight and predation rate (**c**) and clutch weight and lifespan (**d**) across 22 bird sympatric species. All the values are log-transformed and fitted by PGLS methods. Panels show species values (circles) and regression lines with 95% confidence interval (shaded area) are fitted by the PGLS models.

**Table 1 biology-12-01303-t001:** Results of PGLS model to test whether egg mass trade-off against clutch size after controlling for the allometric effects of body mass.

Relationship	*Slope* ± SE	*t*	*p*
Intercept	−1.632 ± 0.309	−5.278	<0.001
Clutch size	−0.123 ± 0.145	−0.845	0.408
Body mass	0.856 ± 0.040	21.261	<0.001

**Table 2 biology-12-01303-t002:** Results of PGLS models to test whether clutch mass is associated with development period, predation rate and lifespan, separately, after controlling for allometric effect of body mass.

Relationship	*Slope* ± SE	*t*	*p*
Intercept	−3.037 ± 0.546	−5.559	<0.001
Development period	0.953 ± 0.179	5.304	<0.001
Body mass	0.675 ± 0.052	12.887	<0.001
Intercept	0.503 ± 0.258	1.949	0.066
Predation rate	−1.204 ± 0.319	−3.774	<0.001
Body mass	0.721 ± 0.058	12.372	<0.001
Intercept	−0.057 ± 0.258	−0.223	0.826
Lifespan	−0.219 ± 0.147	−1.495	0.151
Body mass	0.873 ± 0.071	12.197	<0.001

**Table 3 biology-12-01303-t003:** Results of PGLS models to test whether egg mass and clutch size are associated with development period, predation rate and lifespan separately, after controlling for allometric effect of body mass.

Relationship	*Slope* ± SE	*t*	*p*
Response: Development time			
Intercept	2.900 ± 0.536	5.410	<0.001
Clutch size	0.655 ± 0.155	4.232	0.001
Egg mass	0.556 ± 0.203	2.745	0.013
Body mass	−0.278 ± 0.195	−1.427	0.041
Response: Predation rate			
Intercept	0.280 ± 0.377	0.743	0.467
Clutch size	−0.366 ± 0.121	−3.027	0.007
Egg mass	−0.181 ± 0.159	−1.132	0.273
Body mass	0.016 ± 0.155	0.103	0.619
Response: Lifespan			
Intercept	2.469 ± 1.496	1.649	0.116
Clutch size	−0.667 ± 0.440	−1.515	0.039
Egg mass	−0.196 ± 0.553	−0.355	0.626
Body mass	0.375 ± 0.535	0.702	0.491

## Data Availability

Not applicable.

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
