# Peer review of "Life-History and Ecological Correlates of Egg and Clutch Mass Variation in Sympatric Bird Species at High Altitude"

_biology, 2023, doi:10.3390/biology12101303_

Round 1

Reviewer 1 Report

This study explored the factors associated with the interspecific variation in egg and clutch mass among 22 sympatric bird species at a high altitude in Tibet plateau. It took the advantage that the habitats at high altitudes are relatively more homogenous than those at low land, which facilitate interspecific analyses of egg and clutch mass variation while controlling for various confounding effects of biotic/abiotic factors. In the results, significant associations of egg mass and clutch mass with life history traits and ecological factors were detected, which provide important supports for theories in evolutionary biology. I found the paper well structured, with the background introduced logically, and the results shown clearly and discussed thoroughly. So I just have a few minor points for the authors to consider to revise.

1.    Page 2:In the 2nd paragraph, please check if the sentence “Parental care is an important aspect of reproductive investment in birds, and selection for increased reproductive investment may favor correlated selection on aspects of reproductive investment” correctly confers the meaning that you intended to mean.

2.    Page 4: In the third row of the 2nd paragraph, the sentence should probably be “… following adults with nest material and food to a nest during the incubation and nestling stages”.

3.    Page 4: In the 5th row of the 2nd paragraph, “… by a GPS receiver”.

4.    Page 4: Egg mass was said to be measured before clutch completion. Please provide more details. When there were several eggs in a nest, would the egg mass be averaged?

5.    Page 4: It’s said some species’ maximum longevity were estimated from closely related species. Please indicate in the text about how many species’ longevity was estimated.

6.    Page 5: In the 6th row of the 1st paragraph of the Results, “included as a covariate”.

7.    Page 6: Regarding the models in table 2, please explain why not using a single model with clutch mass as the response variable and development period, predation rate, life span and body mass as the explanatory variables.

The language is overall good and only minor polishment is needed.

Reviewer 2 Report

The authors present a relatively long-term data set of breeding parameters of 22 high-altitude, sympatric, avian species. The paper is well written, and Prof. Peter Lowther has been acknowledged for this. The story flows well, and the analyses are self-explanatory.

However, I have several concerns about the study itself.

The research group appears to be very prolific with the data collected and a considerable part of the literature cited (ca. 25%) are their previous publications. That is a high rate of self-citation. This paper is a collation of these papers and probably other unpublished species. One is directed to get further details (Detailed field procedures for these measures are available in Li et al. (2012, 2015, 2018a)) to these previous publications but should also be a succinct part of this manuscript.

I must admit, as one who has conducted Passerine breeding studies, that the methods applied are extremely disturbing (Pg 4 - Nests were located by flushing the incubating individuals …., a hole dug at the side of the nest chamber ……, ….. frequency of checks was increased to every 1-2 days.) and despite the claim that “no apparent adverse effects were observed; Lu et al. 2011, Li et al. 2015b, Li et al. 2018a” I am very concerned about the field ethics of this study. I also could not find any declaration of appropriate research permits, ringing permits for handling the birds, or other such assurances that the study was supervised properly.

I found this sentence confusing: For each species, clutch mass is calculated as egg mass multiply clutch size. – So was not within-clutch variability taken into consideration? But this is what you discuss in the Introduction.

Table S1. Note that some Latin names are not in Italics. Also of concern is that most species included in the “Current Study” have small sample sizes over the whole decade of study. Is this representative? I am not convinced that these need to be included. Also, the fact that you mix diurnal and nocturnal raptors with Passerines is throwing in variables that are not desirable.

You mention that  - The primary cause of nest failure in bird species is predation ….. but you have ignored the literature on how researchers disclose or lead predators to the studied nests. You must include this concern in the discussion and tell us how you prevented this.

Round 2

Reviewer 2 Report

The paper pertains to high-altitude clutches of a wide range of avian species. The authors have undertaken an extensive review of the draft to answer the reviewer's concerns. I am happy with the answers, modifications, and corrections made by the authors to the issues and concerns I had raised earlier.

There are still language glitches which I suspect will be ironed out in the following editoial handling of the paper.